# Production of Late Seedlings of Açai (*Euterpe oleraceae*) in an Aquaponic System with Tambaqui (*Colossoma macropomum*, Curvier, 1818)

Edileno Tiago de Sousa Nascimento [1], Raimundo Formento Pereira Junior [1], Valéria Silva dos Reis [1], Bianca de Jesus Figueiredo Gomes [1], Marco Shizuo Owatari [2,*], Ronald Kennedy Luz [3], Nuno Filipe Alves Correia de Melo [1], Maria de Lourdes Souza Santos [1], Glauber David Almeida Palheta [1] and Fabio Carneiro Sterzelecki [1]

1   Avenida Presidente Tancredo Neves, Programa de Pós-Graduação em Aquicultura e Recursos Aquáticos Tropicais, Instituto Socioambiental e dos Recursos Hídricos, Universidade Federal Rural da Amazônia, Nº 2501 Bairro, Belém 66077-830, Brazil; edilenotiago2@gmail.com (E.T.d.S.N.); formento.junior1012@gmail.com (R.F.P.J.); valeriareis150@gmail.com (V.S.d.R.); biancajfgomes@gmail.com (B.d.J.F.G.); nuno.melo@ufra.edu.br (N.F.A.C.d.M.); lourdes.santos@ufra.edu.br (M.d.L.S.S.); glauber.palheta@ufra.edu.br (G.D.A.P.); fabio.sterzelecki@ufra.edu.br (F.C.S.)
2   Aquatic Organisms Health Laboratory—AQUOS, Aquaculture Department, Federal University of Santa Catarina (CCA/UFSC), Rodovia Admar Gonzaga 1346, Florianópolis 88040-900, Brazil
3   Laboratório de Aquacultura, Departamento de Zootecnia de Universidade Federal de Minas Gerais (UFMG), Belo Horizonte 31270-901, Brazil; luzrk@vet.ufmg.br
*   Correspondence: marco.owatari@posgrad.ufsc.br

**Abstract:** Over the years, aquaponics has become a powerful technological tool that allows the sustainable production of food, integrating conventional fish farming with vegetable production. The present study evaluated the production of late seedlings of açai *Euterpe oleraceae* in an aquaponic system with tambaqui *Colossoma macropomum*. A total of 36 tambaquis with an initial average weight and length of 1086.75 ± 16.38 g and 38.49 ± 0.90 cm were distributed in 12 independent aquaponic units, totaling 3.62 kg m$^{-3}$. The fish were fed three times daily with commercial feed at a rate of 3%. Three flooding levels of 5, 10 and 15 cm, with constant water flow through the hydroponic bed (0.5 m$^2$), were evaluated, and a control-hydroponic bed with flooding levels of 10 cm was established, all in triplicate. In the 5 and 10 cm treatments, 3450 açai seedlings with an initial height of 12.3 ± 1.9 cm were used (575 per aquaponic units), while the 15 cm treatment contained non-germinated açai seeds. The control group did not receive açai seeds and remained empty. Analyses to monitor total dissolved solids (TDS), electrical conductivity, dissolved oxygen, temperature, pH, alkalinity, hardness, nitrogenous compounds, and phosphate levels were performed. At the end of the 30-day trial, the growth performance of tambaqui and plants was evaluated. Water quality was significantly ($p < 0.05$) affected by different flooding levels. Electrical conductivity and TDS decreased with an increasing in flooding levels. The flooding levels significantly influenced ($p < 0.05$) the concentration of total ammonia and nitrate between the treatments. The 5 cm flooding level showed the best plant development indexes for total height, aerial portion height, root height and aerial portion fresh mass. No significant differences ($p < 0.05$) were observed in tambaqui growth performance. The aquaponic system proved to be effective in reusing fish waste (excreta and feed leftovers) generated in the system. The biotransformation of waste into nutrients allowed the growth of plants and nitrifying bacteria, which, through their metabolic pathways, ensured the purification and reuse of water, avoiding the discharge of this waste into the environment.

**Keywords:** sustainability; regenerative; aquaculture; fish farming; agriculture

## 1. Introduction

Aquaponics is a technological model that harmoniously integrates conventional fish farming with vegetable production in a hydroponic environment, allowing greater food productivity in a sustainable way [1]. In this symbiotic environment, vegetable growth is optimized by the abundant availability of nutrients, generated from the aquaculture system [2,3], which benefits as plants absorb and fix nutrients, contributing to the proper maintenance of water quality parameters in fish ponds [4]. This technology provides a considerable reduction in the use of chemical fertilizers for plant growth, while reducing the environmental impacts generated by fish farming [5,6]. In this way, aquaponics emerges as a circular bioeconomy model for production of organic food [7].

The cultivation techniques most used in aquaponic systems are the Nutrient Film Technique (NFT), the Deep-Water Culture system (DWC) and the media-based grow bed (MGB) [8,9], where various organic, inorganic, or synthetic materials are used as fixation substrates [9], in order to provide greater stability to plant roots, and at the same time, act as a biological support for nitrifying bacteria [1]. This layout establishes both physical and biological filtration for the system, without needing an external biofilter as in other hydroponic setups [10–12]. The use of efficient and low-cost filter media is essential for the improvement of aquaponic systems [13,14]. Given this premise, the substrate with açai seeds (*Euterpe oleraceae* Mart, 1824) can be highlighted, which, in addition to functioning efficiently as an alternative biofilter, produces açai seedlings, generating income for producers [15].

Açaí is a popular Amazonian fruit. In the year 2021 alone, Brazilian production reached a volume close to 227 thousand tons [16], where the state of Pará became a reference in açai production, with an estimated profit of USD 9.5 million for producers in recent years due to exports to the United States [17]. However, the production of açai generates a high volume of fibrous residues, which are often discarded irregularly after processing the fruit, representing a major environmental problem [18,19].

The species reproduces sexually or asexually and the seeds germinate quickly and constantly. Traditionally, the açai seed is sown directly into plastic bags to germinate and form seedlings. This process lasts from 3 to 11 months, so that the plants reach a height of approximately 40 and 50 cm [20]. Despite this, some seeds do not germinate, so it is recommended to sow them in specific containers for germination, and when the seedlings reach the "toothpick" stage, they are transplanted into seedling bags. However, the great challenge is to increase the multiplication rate of vegetative tillers [21]. Thus, the formation of seedlings is recommended in order to obtain a gain of 2 to 3 years in development, in the field, compared to direct sowing [20].

Recently, a pilot study enabled the production of "toothpick" seedlings in aquaponic systems, developing and forming more than 1300 seedlings per m$^2$ in four weeks [15]. In the aforementioned study three flooding levels with constant water flow through the açai seed hydroponic bed were tested. Nevertheless, there are no studies that have evaluated the development of açai seedlings beyond four weeks in an aquaponic system, observing the effects of more robust açai seedlings in these systems.

For this reason, here in the present research, the açai seedlings that were germinated in four weeks in the study proposed by Sterzelecki et al. [15] were used to evaluate the production of late seedlings of açai *Euterpe oleraceae* in aquaponic system with tambaqui *Colossoma macropomum*, verifying the influence of seedlings on water quality, nitrogenous compounds, phosphate levels and on tambaqui growth performance.

## 2. Material and Methods

The experiment was carried out at the Federal Rural University of the Amazon (UFRA)—campus Belém, Pará, Brazil, in an aquaponic system sheltered by a greenhouse, and lasted 30 days. This study was approved by the Ethics Committee on Animal Use of the Federal Rural University of Amazonia protocol number n° 1457260820.

### 2.1. Experimental Design

The experiment was structured with 12 independent aquaponic units, featuring a recirculating water system. Each unit consisted of an individual fish tank with a volume of 1000 L (900 L useful), a 70 L decanter, a 100 L biofilter, a pump (3000 L h$^{-1}$) for water distribution in the system, and a 150 L hydroponic bed filled with açai seeds (13 cm deep) (Figure 1). Dechlorinated water was used to fill the tanks and replace the evaporated water throughout the experiment.

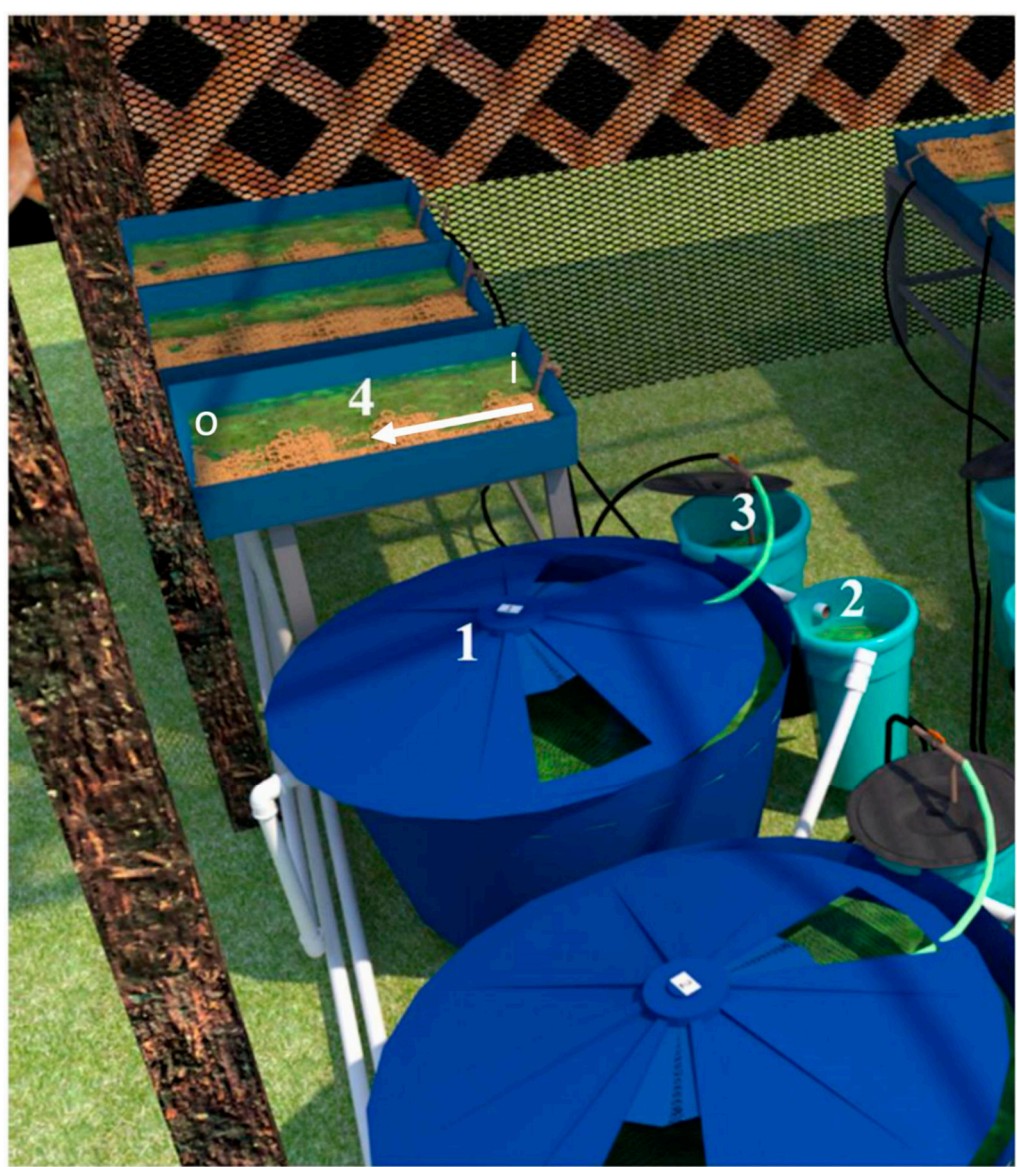

**Figure 1.** Graphical representation of the experimental unit composed of independent aquaponic systems. In (1) 1000 L individual fish tank, (2) decanter—70 L, (3) 100 L biofilter, (4) hydroponic bed—150 L. i = water inlet; o = water outlet.

The fish used in this study came from the Universidade Federal Rural da Amazônia and were obtained in the same spawning from a breeding stock from the Amazonian Aquaculture Biosystems Laboratory. A total of 36 tambaquis with average initial weight and length of 1086.75 ± 16.38 g and 38.49 ± 0.90 cm were used. The tanks were populated with three animals per aquaponic unit (nine per treatment), making a low density of 3.62 kg m$^{-3}$. The fish were fed three times daily with a comercial feed *NUTRIPISCIS*® with a granulometry of 6–8 mm (28% crude protein and 9% lipid) at a rate of 3%.

After the initial procedures for the operation of the aquaponic system, water circulation in the hydroponic beds was allowed. Three flooding levels of 5, 10 and 15 cm with constant flow through the hydroponic bed were tested, and a control-hydroponic bed with flooding level of 10 cm was established, all in triplicate. In the 5 and 10 cm treatments, 3450 açai seedlings (575 per hydroponic bed) with an initial height of 12.3 ± 1.9 cm were used (Figure 2), while the 15 cm treatment contained non-germinated açai seeds during the study by Sterzelecki et al. [15]. The control group did not receive açai seeds and remained empty according to Sterzelecki et al. [15].

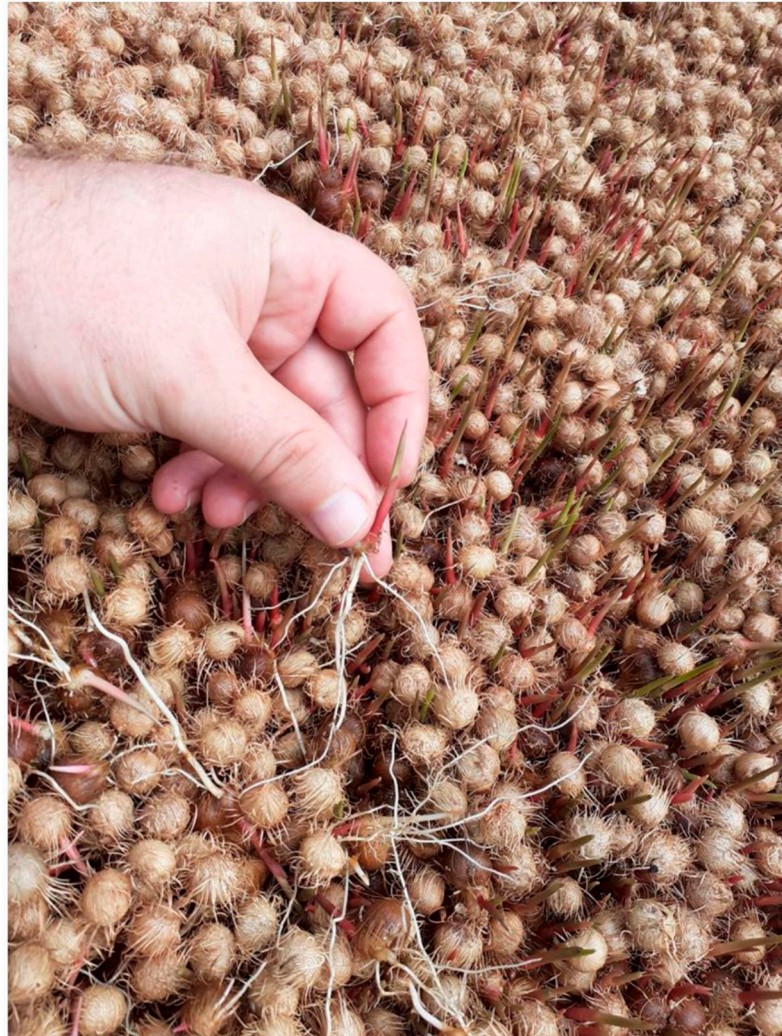

**Figure 2.** Acai seedlings obtained by Sterzelecki et al. [15] in aquaponic system with tambaqui *Colossoma macropomum* and used in the present study, giving experimental continuation for production of late seedlings. The figure shows germinated and non-germinated seeds of Açai palm *Euterpe oleracea*.

## 2.2. Water Analysis

Daily analyses were carried out in the fish tanks to monitor total dissolved solids (TDS) (AQUAREAD AP-800 Multiparameter Probe), electrical conductivity in the water, dissolved oxygen (YSI ProODO, Yellow Springs, OH, USA, ±0.01 mg L$^{-1}$), temperature and pH (BL-1072—portable digital pH meter). In addition, dissolved oxygen and pH were also analysed in influent and effluent water from the hydroponic beds.

Water samples were collected weekly to check alkalinity, hardness, nitrogenous compounds and phosphate levels. The water was filtered using a 0.7 μm F/GF membrane and analyzed based on standard APHA methods [22]. Phosphate levels were measured by applying the test for total phosphorus (ascorbic acid) [22], total ammonia [23], nitrite with the Griess reaction [22], and nitrate with spectrophotometry read at 220 nm/270 nm in a spectrophotometer (Ionlab, Paraná, Brazil) [22]. Total ammonia removal (%) was calculated as: TAN removed (%) = (TAT − TAC/TAC) × 100, where TAN is the total ammonia, TAT is the total ammonia in the treatment tank, and TAC is the ammonia total in the control tank.

### 2.3. Fish Growth Performance

Fish were measured and weighed weekly to verify growth performance. Before sampling, individuals were fasted for 24 h and anesthetized with Eugenol (50 mg $L^{-1}$) to avoid stress. Weight gain was obtained by subtracting the final weight from the initial weight. Feed conversion rates were calculated by (FCR) = feed intake (g)/weight gain (g). Fulton's condition factor (K) = $W/L^3 × 100$.

### 2.4. Plant Growth Performance

The performance of the açai seedlings was verified at the end of 30 days through the development in relation to the total height of the plant (cm), aerial portion height (cm), root height (cm), collar diameter (mm), total fresh mass (g), aerial portion fresh mass (g), and root fresh mass (n = 30 per treatment).

### 2.5. Statistical Analysis

The homoscedastic and normality dispersion of the data was analyzed. For parametric variables, post hoc one-way ANOVA and Tukey tests were used to verify significant differences ($p < 0.05$). For non-parametric results, post hoc Kruskal–Wallis and Dunn tests were applied to explore significant differences ($p < 0.05$). Two-way ANOVA was used on nitrogen and phosphate compounds to compare different sampling times ($p < 0.05$). GraphPad Prism 9 was used for statistical analysis.

## 3. Results

### 3.1. Water Quality

Water quality variables were significantly affected ($p < 0.05$) by different flooding levels. The flooding levels significantly influenced ($p < 0.05$) the concentration of total ammonia and nitrate between treatments. Total ammonia was significantly reduced throughout the experimental period in groups with seedlings and açai seeds, regardless of the flooding level. The lowest nitrate concentrations were observed in the treatment with a flooding level of 15 cm. However, nitrate was lower in the flooding level 15 cm only in the 21st day, when compared to the other treatments. Nitrite and phosphate were not significant between groups (Table 1). Dissolved oxygen showed a significant difference ($p < 0.05$) between the control group and the flooding level of 10 cm. The pH was depth-dependent on the flooding levels, while the conductivity and amount of TDS decreased with increasing depth of the system (Table 2).

### 3.2. Plant Growth Performance

The flooding levels of 5 and 10 cm significantly influenced plant performance. The 5 cm flooding level showed the best development indexes in relation to total height, aerial portion height, root height and aerial portion fresh mass. However, collar diameter, total fresh mass and root fresh mass did not differ between the groups. The treatment with a depth of 15 cm did not show germinated açai seedlings, while the control treatment was empty and did not have açai seedlings (Table 3).

**Table 1.** Effects of seeds and late seedlings of açai *Euterpe oleraceae* in aquaponic system with tambaqui *Colossoma macropomum* on nitrogen and phosphate compounds at different flooding levels (5, 10 and 15 cm), and control group. Data are presented as the mean ± SEM. Different letters indicate statistical differences between the groups ($p < 0.05$).

| Compounds | Days | Control | 5 cm | 10 cm | 15 cm |
|---|---|---|---|---|---|
| Total ammonia (mg L$^{-1}$) | 0 | 7.73 ± 2.86 [a] | 7.73 ± 2.86 [b] | 0.45 ± 0.08 [b] | 0.14 ± 0.03 [b] |
| | 14 | 15.76 ± 14.77 [a] | 0.97 ± 0.67 [b] | 0.47 ± 0.18 [b] | 0.75 ± 0.82 [b] |
| | 21 | 13.55 ± 14.11 [a] | 0.26 ± 0.21 [b] | 0.17 ± 0.06 [b] | 0.10 ± 0.03 [b] |
| | 28 | 10.07 ± 12.44 [a] | 0.39 ± 0.28 [b] | 0.34 ± 0.16 [b] | 0.34 ± 0.18 [b] |
| Nitrite (mg L$^{-1}$) | 0 | 0.00 ± 0.00 | 0.00 ± 0.00 | 0.00 ± 0.00 | 0.00 ± 0.00 |
| | 14 | 0.03 ± 0.00 | 0.02 ± 0.00 | 0.02 ± 0.01 | 0.03 ± 0.01 |
| | 21 | 0.03 ± 0.02 | 0.02 ± 0.01 | 0.01 ± 0.01 | 0.01 ± 0.02 |
| | 28 | 0.01 ± 0.00 | 0.01 ± 0.02 | 0.02 ± 0.01 | 0.02 ± 0.00 |
| Nitrate (mg L$^{-1}$) | 0 | 26.45 ± 4.70 [a] | 24.57 ± 0.10 [a] | 18.78 ± 12.33 [ab] | 2.14 ± 0.63 [b] |
| | 14 | 36.16 ± 0.54 [a] | 33.85 ± 4.33 [a] | 20.40 ± 14.44 [ab] | 7.83 ± 0.47 [b] |
| | 21 | 33.99 ± 3.31 [a] | 30.08 ± 9.46 [ab] | 15.25 ± 15.04 [b] | 7.23 ± 0.07 [c] |
| | 28 | 30.15 ± 6.30 [a] | 29.31 ± 7.28 [a] | 17.02 ± 9.41 [ab] | 6.55 ± 1.69 [b] |
| Phosohate (mg L$^{-1}$) | 0 | 5.63 ± 1.30 | 6.23 ± 0.80 | 6.06 ± 0.39 | 6.00 ± 0.21 |
| | 14 | 7.77 ± 0.08 | 7.74 ± 0.27 | 6.48 ± 1.28 | 6.76 ± 0.61 |
| | 21 | 7.37 ± 0.77 | 7.60 ± 0.25 | 7.29 ± 0.56 | 6.77 ± 0.10 |
| | 28 | 6.89 ± 0.47 | 6.54 ± 0.38 | 6.61 ± 1.33 | 6.08 ± 0.96 |

**Table 2.** Effects of seeds and late seedlings of açai *Euterpe oleraceae* in aquaponic system with tambaqui *Colossoma macropomum* on water quality at different flooding levels (5, 10 and 15 cm), and control group. Data are presented as the mean ± SEM. Different letters indicate statistical differences between the groups ($p < 0.05$).

| Variables | Flooding Levels | | | |
|---|---|---|---|---|
| | Control (10 cm) | 5 cm | 10 cm | 15 cm |
| Temperature °C | 27.7 ± 0.06 [a] | 27.8 ± 0.06 [a] | 27.6 ± 0.07 [ab] | 27.4 ± 0.06 [b] |
| Dissolved oxygen (mg L$^{-1}$) | 5.37 ± 0.09 [a] | 5.13 ± 0.08 [ab] | 4.98 ± 0.10 [b] | 5.2 ± 0.09 [ab] |
| pH | 6.4 ± 0.05 [c] | 6.85 ± 0.05 [b] | 6.99 ± 0.04 [b] | 7.16 ± 0.03 [a] |
| Electrical conductivity (μS cm$^{-1}$) | 423.2 ± 12.30 [a] | 344.0 ± 7.87 [b] | 328.9 ± 8.14 [b] | 266.5 ± 8.20 [c] |
| Total dissolved solids (mg L$^{-1}$) | 276.7 ± 7.70 [a] | 226.6 ± 5.80 [b] | 206.4 ± 6.00 [b] | 180.0 ± 7.61 [c] |

**Table 3.** Development parameters of açai seedlings after 30 days in aquaponic system at different flooding levels (5, 10 cm). Data are presented as the mean ± SEM based on sampling of 15 plants per aquaponic bed (45 per treatment). Different statistical letters indicate differences between the groups ($p < 0.05$).

| Development Parameters | Flooding Levels | |
|---|---|---|
| | 5 cm | 10 cm |
| Plant initial height (cm) | 12.3 ± 1.9 [a] | 12.3 ± 1.9 [a] |
| Plant total height (cm) | 34.55 ± 0.72 [a] | 24.7 ± 0.57 [b] |
| Aerial portion height (cm) | 20.61 ± 0.51 [a] | 13.41 ± 0.40 [b] |
| Root height (cm) | 13.93 ± 0.40 [a] | 11.29 ± 0.33 [b] |
| Collar diameter (mm) | 3.4 ± 0.071 | 3.56 ± 0.061 |
| Total fresh mass (g) | 1.88 ± 0.03 | 1.83 ± 0.04 |
| Aerial portion fresh mass (g) | 0.87 ± 0.03 [a] | 0.66 ± 0.02 [b] |
| Root fresh mass (g) | 0.98 ± 0.03 [b] | 1.19 ± 0.04 [a] |

### 3.3. Fish Growth Performance

No significant differences ($p < 0.05$) were observed in tambaqui growth performance indexes, regardless of the flooding levels tested. Furthermore, no fish mortalities were observed during the experimental period (Table 4).

**Table 4.** Effects of seeds and late seedlings of açai *Euterpe oleraceae* at different flooding levels (5, 10 and 15 cm) on the growth performance of tambaqui *Colossoma macropomum,* and control group. Data are presented as the mean $\pm$ SEM n = 9.

| Growth Indexes | Flooding Levels | | | |
| --- | --- | --- | --- | --- |
| | Control (10 cm) | 5 cm | 10 cm | 15 cm |
| Initial weight (g) | 1097.0 $\pm$ 50.0 | 1103.0 $\pm$ 57.6 | 1080.0 $\pm$ 56.3 | 1067.0 $\pm$ 54.8 |
| Initial condition factor | 1.83 $\pm$ 0.12 | 1.95 $\pm$ 0.14 | 1.78 $\pm$ 0.13 | 2.05 $\pm$ 0.16 |
| Final weight (g) | 1388.0 $\pm$ 69.6 | 1427.0 $\pm$ 76.6 | 1396.0 $\pm$ 71.1 | 1396.0 $\pm$ 65.1 |
| Weight gain (g) | 291.3 $\pm$ 31.3 | 324.0 $\pm$ 35.9 | 315.6 $\pm$ 28.3 | 329.4 $\pm$ 36.4 |
| Initial length (cm) | 39.12 $\pm$ 0.6 | 38.33 $\pm$ 0.7 | 39.26 $\pm$ 0.5 | 37.28 $\pm$ 0.6 |
| Final length (cm) | 41.02 $\pm$ 0.9 | 42.6 $\pm$ 0.6 | 41.92 $\pm$ 0.7 | 41.8 $\pm$ 0.7 |
| Final condition factor | 2.01 $\pm$ 0.11 | 1.84 $\pm$ 0.6 | 1.89 $\pm$ 0.12 | 1.91 $\pm$ 0.14 |
| Feed conversion ratio | 2.07 $\pm$ 0.16 | 1.87 $\pm$ 0.18 | 1.87 $\pm$ 0.12 | 1.80 $\pm$ 0.22 |

### 4. Discussion

Aquaponics has all the fundamental characteristics established as basic criteria for a more sustainable future in food production, and meets the principles of circular bioeconomy, being able to optimize the efficiency of resources used in production and mitigate the environmental impacts caused by conventional aquaculture [1,7].

As observed by Sterzelecki et al. [15], in the present study the use of seed and late seedlings of açai in different flooding levels produced significant positive effects on water quality and plant development in the long term. The water quality in an aquaponic system can be affected mainly by the stocking density of fish, plants and microbiological activities [24], in addition to providing an increase in productivity in the system when maintained in optimal conditions [25–28]. This information corroborates the findings of the present study, where a lower availability of dissolved oxygen was observed at flooding levels of 10 cm, which possibly resulted in lower plant productivity rates.

Likewise, the pH variation recorded here can be related to the characteristics of the aquaponic sets presented. In the control group, which contained only the fish tank without the açai bed, the pH remained significantly more acidic. This probably occurred due to the absence of plants in this treatment, which reduced the ability to remove nitrogenous compounds from the water, increasing the levels of N-NH$_3$ and Carbon dioxide (CO$_2$) in the aqueous medium, and consequently, the acidity of the water. While in the other treatments that had açai seedlings or seeds, the pH remained close to neutrality, possibly providing a greater proliferation of nitrifying or heterotrophic bacteria, which have optimized growth at a pH between 7.0 and 8.0 [29]. Despite the values found in the present study being a little below the level indicated for plants and fish, no apparent harmful effects were observed on the development of açai seedlings and fish, indicating that these conditions were acceptable for the aquaponic system proposed in the present study.

Electrical conductivity is a good indicator of ion availability for plants [30], as it varies considerably according to the concentration of dissolved salts in water [31]. The results found in the present study demonstrate that the açai seedlings efficiently absorbed the nutrients available in the water [7], significantly reducing the electrical conductivity in the treatment with flooding level of 5 cm, where the plants ended the experiment with greater total height, implying a greater demand and absorption of nutrients from water. However, high flooding levels are not favorable for the system [15], given that the absorption of nutrients available within reach of plant roots is limited when the plant is in an early stage of development.

Total dissolved solids at elevated levels can significantly affect fish and aquatic life in general, and are extremely important indicators of deteriorating water quality and degradation of aquatic environments [32]. In turn, plant roots have the ability to filter such particles dissolved in water and use them as nutrients, alleviating the amount of total dissolved solids in a given aquatic ecosystem [33]. Here, in the present study, we didactically observed such an event, where the control treatment had higher concentrations of TDS, probably because it did not have plants in its configuration. While the groups with flooding levels 5 and 10 cm significantly reduced the amount of TDS. Interestingly, the group with flooding levels at 15 cm, which contained only açaí seeds, had the lowest amount of TDS, demonstrating that açaí seeds are also an excellent medium for biological support, capable of removing TDS.

Nitrogen compounds, in their various forms, can be removed from cropping systems through mechanical, physical-chemical and biological processes. However, biological processes prove to be more economical and efficient, as they follow the same decomposition pathways that exist in nature. In effluent treatment systems, the main biological processes for nitrogen cycling are nitrification, denitrification and anaerobic ammonia oxidation [15,34–36]. In aquaponics, the nitrification process takes place in two stages. In the first step, called nitritation, ammonia is oxidized to nitrite by ammonia-oxidizing bacteria (*Nitrosomonas* and *Nitrosospir*). Then, in the phase known as nitratation, nitrite is oxidized to nitrate by nitrite-oxidizing bacteria (*Nitrosococcus*, *Nitrobacter*, *Nitrospira*, *Nitrococcus* and *Nitrospina*) [36,37]. However, the natural establishment of nitrifying bacterial communities is slow [36].

In the present study, it was observed that the treatment with flooding levels at 15 cm, which contained only açaí seeds, probably offered greater surface area of biological support for the adhesion of nitrite-oxidizing bacteria, given the kinetics of degradation of this nitrogenous compound observed on the 21st day (Table 1). Nevertheless, the groups in flooding levels at 5, 10 and 15 cm were efficient in the oxidation of total ammonia when compared to the control treatment, possibly due to the greater availability of biological support for the adhesion of ammonia-oxidizing bacteria (*Nitrosomonas* and *Nitrosospir*), considering the absence of plants and seeds in the control group. Even so, we observed that in the groups of flooding levels at 5, 10 cm, the total ammonia remained low since the beginning of the period, indicating that the reduction in nitrogenous compounds in this period also occurred through the absorption of the açaí seedlings roots, while the presence of phototrophic microorganisms may justify the reduction in nitrate in flooding levels at 15 cm, as already demonstrated by Sterzelecki et al. [15]. Such results demonstrate that the açaí seed acted efficiently within the aquaponic system, significantly improving the water quality, and can be considered an excellent live biofilter for the system.

Plant development can be significantly influenced according to the depth level where it is found [38], and depending on the water depth, the survival capacity of plants can decrease [39]. Although the açaí tree presents adaptations to flooding situations [40], the results obtained in the present study indicate that the flooding level of 5 cm presented the best results for the analyzed parameters. Similar results related to the development of the aerial portion of açaí seedlings in an aquaponic system were also found by Medina et al. [41] when comparing the effects of two feedings on the productivity of red amaranth (*Amaranthus tricolor*) in an integrated culture with blue tilapia (*Oreochromis aureus*) during 60 days. At the time, the researchers verified that the increased productivity of the plant, combined with the use of a low-protein fish feed, can increase the total revenue of the aquaponic farm, despite the reduction in fish production.

Although root height was greater at the 5 cm flooding level, fresh root mass was greater at the 10 cm level. This result may be directly related to the depth level and to the açaí substrates, since, due to the greater depth in the 10 cm flooding levels, the plant probably produced more dispersed roots to optimize the absorption of nutrients in the water. Results similar to those of the present study were also found by Fischer et al. [42] when comparing the productivity of juvenile largemouth bass *Micropterus salmoides* in a

recirculating aquaculture system versus aquaponics with lemongrass and spring onion production. At the time, the authors considered *M. salmoides* as a suitable fish species to grow with aquaponics.

Tambaqui *C. macropomum* is considered a rustic species that grows best in slightly acidic waters [43], as do most plants [24], which results in two essential factors for integrated agriculture. Despite this, no improvements were observed in the growth performance of the fish in the present study. Similar results to those presented here were described by Silva et al. [44] integrating the tambaqui cultivation with lettuce *Lactuca sativa* production; by Da Costa et al. [45] integrating the tambaqui with coriander *Coriandrum sativum* seedlings production; by Araújo et al. [46] integrating the tambaqui with Italy tomato *Solanun lycopersicum* production, and by Sterzelecki et al. [15] integrating the tambaqui with açai seedlings production, demonstrating a pronounced potential of the species for production in aquaponic systems.

In the present study, the use of a low-cost medium in aquaponics proved to be effective in reusing fish waste (excreta and feed remains) generated in the system. The biotransformation of waste into nutrients in the aquatic environment allowed the growth of hydroponic plants and nitrifying bacteria which, through their metabolic pathways, ensured that the water was purified and reused for fish growing, preventing the discharge of this waste into the environment.

## 5. Conclusions

Under the conditions proposed in the present study, the açai seed biofilter maintained the water quality variables within the range tolerated by tambaqui, regardless of the flooding levels used. For better production of late seedlings of açai *E. oleraceae* in aquaponic system with tambaqui *C. macropomum*, a flooding level of 5 cm is recommended, where the best results of plant performance were observed, without negatively affecting water quality and fish growth. Obtaining late seedlings of açai in an aquaponic system proved to be efficient and easy to handle, allowing sowing and production of vegetable seedlings in the same place. We recommend further studies to verify the economic feasibility of producing açai seedlings in aquaponics.

**Author Contributions:** Experimental execution, methodology, investigation, data curation, and writing—original draft, E.T.d.S.N.; experimental execution and investigation, R.F.P.J., V.S.d.R. and B.d.J.F.G.; writing—review and editing, M.S.O.; funding acquisition, conceptualization, and methodology, R.K.L., N.F.A.C.d.M. and M.d.L.S.S.; funding acquisition, conceptualization, methodology, validation, and supervision, G.D.A.P.; funding acquisition, conceptualization, methodology, validation, supervision, and writing—review and editing, F.C.S. All authors have read and agreed to the published version of the manuscript.

**Funding:** This research was funded by Coordenação de Aperfeiçoamento de Pessoal de Nível Superior—CAPES for partially funding this research grant number [001], by the PROCAD Amazônia grant number [21/2018], and the APC was funded by Ronald Kennedy Luz (CNPq n°. 308547/20187).

**Institutional Review Board Statement:** This study was approved by the Ethics Committee on Animal Use of the Federal Rural University of Amazonia protocol number n° 1457260820.

**Data Availability Statement:** Data will be made available on request.

**Acknowledgments:** The authors would like to thank the Coordenação de Aperfeiçoamento de Pessoal de Nível Superior—CAPES for partially funding this research, the PROCAD Amazônia for financial support to Nuno F. A. Correia de Melo, 40221741291, public notice n° 21/2018, project n° 88887.200588/2018-00, and thank the Conselho Nacional de Desenvolvimento Científico e Tecnológico—CNPq Brasil (Project n° 402952/ 2021-9) for financial support to Ronald Kennedy Luz (CNPq n°. 308547/20187).

**Conflicts of Interest:** The authors declare that they have no known competing financial interest in this manuscript.

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
