# Peer review of "Production of Late Seedlings of Açai (Euterpe oleraceae) in an Aquaponic System with Tambaqui (Colossoma macropomum, Curvier, 1818)"

_agriculture, doi:10.3390/agriculture13081581_

Round 1

Reviewer 1 Report

English language needs to be improved 

Author Response

Dear Moira Liu Editor and Reviewers

We really appreciated the suggestions and corrections on the manuscript, certainly, these suggestions improved the understanding of the article beyond its quality. We are resubmitting two versions of the manuscript, one of them with the changes highlighted in yellow and one with the changes already made (clean version - manuscript mandatory). Please find below the main corrections already performed.

REVIEWER REPORTS

Reviewer Comments:

Reviewer 1

The introduction section in scarce and does not include a proper state of the art for the approached subject. The authors should address the literature and highlight the importance, the need and the originality of their study, compared to other similar studies.

Answer: We appreciate all corrections and raised points. Surely, they will greatly enrich the present work. We include more information in the introduction between lines 95~103 and 107~109.

Line 31 – 32 – What type of plant performance? Do you mean plant growth performance? Please add quantitative data to support your statement.

Answer: Thank you reviewer. We apologize for the misunderstanding. The sentence has been changed to: (Lines 53~55) The 5 cm flooding level showed the best plant development indexes for total height, aerial portion height, root height and aerial portion fresh mass.

Line 33 – Please ass the value of p in brackets to support you statistical statement.

Answer: Thank you the reviewer. Thank you for emphasizing these points. We appreciate the reviewer's accurate observation. Was inserted.

Line 34 - ,,the aquaponic system”

Answer: Thank you, reviewer. We apologize for the misunderstanding. The sentence has been changed.

Line 56 -57 – The authors should mentioned other studies which used low-cost media for aquaponics purpose, such as https://doi.org/10.3390/plants12030540

Answer: Thank you, the reviewer. Thank you for emphasizing these points.

The reference was inserted in lines 86~87.

Line 91 – What type of hydraulic regime was applied in the hydroponic units? It is very important it should be added to materials and methods section.

Answer: Thank you, the reviewer. Thank you for emphasizing these points. We appreciate the reviewer's accurate observation. Was inserted in the lines 124~125.

Line 97 – The applied stocking density of fish in the rearing units is unusually low for an aquaponic system. Please provide an explanation for applying this density.

Answer: Thank you, the reviewer. Thank you for emphasizing these points. In the present aquaponic system, planting was carried out in a substrate culture environment and requires a low fish density, since it is a production on an experimental scale. These conditions can also be applied to reduce the occurrence of diseases and/or control the environmental conditions for plants used in the aquaponic set. However, we make it clear that it is a low density in the line 133.

Line 105 – Please extend your explanation related to the control group. It is not clear what kind of hydroponic bed was used in the control variant. Also, why did you used non-germinated acai seedlings in the 15 cm treatment?

Answer: Thank you, the reviewer. Thank you for emphasizing these points. We appreciate the reviewer's accurate observation. As mentioned in the objective of the study, we continued the study by Sterzelecki et al. (2022), therefore, we maintained the same design used by them. The control did not contain açai seeds, so that we could observe whether the growth of the fish would be influenced by the presence of seedlings. (2022), were used to verify if they would germinate in long-term.

The sentence was rewritten.

Lines 136~142: After the initial procedures for the operation of the aquaponic system, water circulation in the hydroponic beds was allowed. Three flooding levels of 5, 10 and 15 cm with constant flow through the hydroponic bed were tested, and a control-hydroponic bed with flooding level of 10 cm was established, all in triplicate. In the 5 and 10 cm treatments, açai seedlings with an initial height of 12.3 ± 1.9 cm were used, while the 15 cm treatment contained non-germinated açai seeds during the study by Sterzelecki et al. [14]. The control group did not receive açai seeds and remained empty according to Sterzelecki et al. [14].

Line 125 – Please replace Zootechnical performance with Fish growth performance.

Answer: Thank you, the reviewer. The sentence has been changed.

In this section I suggest to calculate the condition factor (Fulton coefficient) at the beginning of the experiment and at the end of the experiment, according to the following paper ,,GROWTH PERFORMANCE AND CONDITION FACTOROF OREOCHROMIS NILOTICUS SPECIES FEED WITH A DIETWHICH INCLUDE SOME PHYTO-ADDITIVES”. Further, the allometric coefficient needs to be calculated in order to assess the fish growth process.

Answer: Thank you, the reviewer. Thank you for emphasizing this point. We apologize. We did not identify any significant differences in the growth performance of the fish, so we had not entered data related to the condition factor. The data were included as suggested by the reviewer in the Table 3.

Line 130 - ,,Plant growth performance”

Answer: Thank you, the reviewer. The sentence has been changed.

In the RESULTS section, I recommend remaking the graphs using different colors to depict the experimental variants. It is difficult to read in black color. As well, please add a table with threshold values for water quality indicators in aquaponic/RAS systems.

Answer: Thank you, the reviewer. Thank you for emphasizing this point. We apologize.

We change figure 2 for a table 1.

We believe that the water quality indicators may vary according to the species used; for example, maintaining an ideal water temperature between 22 − 24 °C, pH in the range of 5.6 − 7.3 and DO of 3 − 10 mg/L seems good for tilapia and some plants. However, for us to state or indicate what is the limit for water quality values in aquaponics is very complex, given that aquaponics is in constant investigation.

Line 199 – Please avoid using chemical structure (CO2) if you do not provide the prior the full name of the structure. Ex. Carbon dioxide (CO2).

Answer: Thank you, reviewer. The sentence has been changed.

The conclusion section should be rephrased. What is optimal water quality? You did not give quantitative data to support that statement. Also, in terms of fish growth, what does good growth mean? You need to be more specific and indicate data in order to avoid speculation.

Answer: Thank you, reviewer. We apologize for the misunderstanding. The sentence has been changed to:

  1. Conclusion

Under the conditions proposed in the present study, the açai seed biofilter maintained the water quality variables within the range tolerated by tambaqui, regardless of the flooding levels used. For better production of late seedlings of açai E. oleraceae in aquaponic system with tambaqui C. macropomum, a flooding level of 5 cm is recommended, where the best results of plant performance were observed, without negatively affecting water quality and fish growth.

Thank you, the reviewer. Thank you for emphasizing these points. Peer review greatly enriches our work. All these points indicated by the reviewer are relevant.

Sincerely yours,

Owatari, Marco Shizuo

Brazil, July 2023.

Reviewer 2 Report

It took me a while to realize that this was a media system and both the support media and the plants were Acai seed/seedlings. This needs to be stated more clearly. A photo of the harvested fruit, extracted seeds and seedling would be useful to readers not familiar with this plant.

L75-76. Ok, but this study is also not longer than 4 weeks.

The numbers of plants/bed needs to be stated and production expressed in terms of kg/bed. 

The inclusion of the 15 cm treatment with non-germinated acai seeds needs to be better explained. 

While the flooding levels are given, the depth of the media bed is omitted. 

Figure 1. Replace with a simple process flow diagram.

Was the TDS estimated from the EC? Need to define how it was estimated.

Line 122. Total ammonia removal was not discussed in the M.S.

If you are interested total ammonia removal across the plant bed, the TAC should be based on the individual bed values. If you are interested in the total system removal the influent TAN must be based on nitrogen contained in the feed. This is a much more complex computation. 

Line 139-140. Why was the two-way ANOVA restricted to nitrogen and phosphate?

Figure 2. These figures need to be enlarged or converted to tables. In their current state, I cannot distinguish between the different treatments. Zero information content!

Table 1, total dissolved solids for 5 cm. This value is not correct!

Table 2, last three entries (total fresh mass, aerial portion, and root weights). These parameters must be expressed on a bed basis for both the initial and final values.

Lines 216-226. There appears to be confusion between suspended solids (SS) and total dissolved solids (TDS). These are entirely two different parameters. SS is a measure of the solids retained on 0.45 um filter while TDS is a measure of dissolved ions. The reduction in TDS in the 5, 10, and 15 cm treatments may be due to ion exchange uptake by the extracted seeds. Why the lowest values were found in the 15 cm treatment remains to be determined. This could be a simple dilution impact as the 15 cm treatment has a larger water volume.

References

2.                     More information is needed on this reference

15 & 16.         Convert access information to English and standardized format

21                   Date not in correct location and city/country missing 

23.                   Journal and volume needed

29.                   Year not in correct location

43.                   The second e543997543 is the same as the first e543997543

45                   Page number

Reference [44] is not cited in the text.

The English is quite good!

Author Response

Dear Moira Liu Editor and Reviewers

We really appreciated the suggestions and corrections on the manuscript, certainly, these suggestions improved the understanding of the article beyond its quality. We are resubmitting two versions of the manuscript, one of them with the changes highlighted in yellow and one with the changes already made (clean version - manuscript mandatory). Please find below the main corrections already performed.

REVIEWER REPORTS

Reviewer Comments:

Reviewer 2

It took me a while to realize that this was a media system and both the support media and the plants were Acai seed/seedlings. This needs to be stated more clearly. A photo of the harvested fruit, extracted seeds and seedling would be useful to readers not familiar with this plant.

Answer: We appreciate all corrections and raised points. Surely, they will greatly enrich the present work.

A new figure 2 was inserted with details of the seeds and germinated seeds used in the study.

L75-76. Ok, but this study is also not longer than 4 weeks.

Answer: Thank you reviewer. We apologize for the misunderstanding.

The study was a continuation of the study by Sterzelecki et al. (2022) which lasted 30 days. The seedlings generated in the study by Sterzelecki continued for another 30 days in the present study. Thus, they were evaluated for 60 days. It was poorly explained in the manuscript. The sentence has been changed to:

Lines 105~115: Recently, a pilot study enabled the production of “toothpick” seedlings in aquaponic systems, developing and forming more than 1,300 seedlings per m2 in four weeks [15]. In the aforementioned study three flooding levels with constant water-flow through the açai seed hydroponic bed were tested. Nevertheless, there are no studies that have evaluated the development of açai seedlings beyond four weeks in an aquaponic system, observing the effects of more robust açai seedlings in these systems.

For this reason, here in the present research, the açai seedlings that were germinated in four weeks in the study proposed by Sterzelecki et al. [15] were used to evaluate the production of late seedlings of açai Euterpe oleraceae in aquaponic system with tambaqui Colossoma macropomum, verifying the influence of seedlings on water quality, nitrogenous compounds, phosphate levels and on tambaqui growth performance.

The numbers of plants/bed needs to be stated and production expressed in terms of kg/bed.

Answer: Thank you for emphasizing this point. The numbers of plants/bed were inserted.

Line 140: In the 5 and 10 cm treatments, 3450 açai seedlings (575 per hydroponic bed) with an initial height of 12.3 ± 1.9 cm were used...

The inclusion of the 15 cm treatment with non-germinated acai seeds needs to be better explained.

Answer: Thank you for emphasizing this point. As the study was a continuation of the research proposed by Sterzelecki et al. (2022), seeds that did not germinate at the time, were used in the present study to find out whether they would germinate or remain without germinating. The information were inserted in the line 142

While the flooding levels are given, the depth of the media bed is omitted.

Answer: Thank you for emphasizing this point. The the depth of the media bed was inserted in the line 127.

Figure 1. Replace with a simple process flow diagram.

Answer: Thank you, reviewer. Sometimes we, the authors, are in a delicate situation when one of the reviewers wants a change, while the other reviewer wants another. Which of the reviewers to meet? In this way, we believe that keeping figure 1 would be the best option to visualize the experimental design.

Was the TDS estimated from the EC? Need to define how it was estimated.

Answer: Thank you for emphasizing this point. This information is in the material and methods section, between lines 150 and 151.

Line 122. Total ammonia removal was not discussed in the M.S.

Answer: Thank you for emphasizing this point. Nitrogenous compounds were addressed in the lines 245~256 and 277~301.

If you are interested total ammonia removal across the plant bed, the TAC should be based on the individual bed values. If you are interested in the total system removal the influent TAN must be based on nitrogen contained in the feed. This is a much more complex computation.

Answer: Thank you for emphasizing this point. We used this condition for the present study. Lines 160~162: Total ammonia removal (%) was calculated as: TAN removed (%) = (TAT-TAC/TAC) × 100, where TAN is the total ammonia, TAT is the total ammonia in the treatment tank, and TAC is the ammonia total in the control tank.

Line 139-140. Why was the two-way ANOVA restricted to nitrogen and phosphate?

Answer: Thank you for emphasizing this point. Because collections were carried out at different times during the 30 days of experimentation.

Figure 2. These figures need to be enlarged or converted to tables. In their current state, I cannot distinguish between the different treatments. Zero information content!

Answer: Thank you for emphasizing this point. Was converted to table 2.

Table 1, total dissolved solids for 5 cm. This value is not correct!

Answer: We appreciate the reviewer's accurate observation. We correct the observed fault. It was a typing error.

Table 2, last three entries (total fresh mass, aerial portion, and root weights). These parameters must be expressed on a bed basis for both the initial and final values.

Answer: Thank you for emphasizing this point. The data are presented as mean ± SEM based on sampling of 15 plants per aquaponic bed (45 per treatment). The information was inserted in the (now) table 3 caption.

Lines 216-226. There appears to be confusion between suspended solids (SS) and total dissolved solids (TDS). These are entirely two different parameters. SS is a measure of the solids retained on 0.45 um filter while TDS is a measure of dissolved ions. The reduction in TDS in the 5, 10, and 15 cm treatments may be due to ion exchange uptake by the extracted seeds. Why the lowest values were found in the 15 cm treatment remains to be determined. This could be a simple dilution impact as the 15 cm treatment has a larger water volume.

Answer: Thank you reviewer. We apologize for the misunderstanding. TDS was measured in the fish tank and the water volume in the system was identical in all groups. The flooding levels (5, 10, 15 cm) in the beds was regulated by a drainage siphon. The sentence has been rewritten. In lines 266~275.

References

  1. More information is needed on this reference

Answer: Thank you reviewer. Was inserted.

15 & 16.         Convert access information to English and standardized format

Answer: Thank you reviewer.

21                   Date not in correct location and city/country missing

Answer: Thank you reviewer. Was inserted.

  1. Journal and volume needed

Answer: Thank you reviewer. This open access book, written by world experts in aquaponics and related technologies, provides the authoritative and comprehensive overview of the key aquaculture and hydroponic and other integrated systems, socio-economic and environmental aspects. Publisher Springer Nature.

  1. Year not in correct location

Answer: Thank you reviewer. Was inserted.

  1. The second e543997543 is the same as the first e543997543

Answer: Thank you reviewer. Was corrected.

45                   Page number

Answer: Thank you reviewer. Was corrected.

Reference [44] is not cited in the text.

Answer: Thank you reviewer. Was corrected.

Sincerely yours,

Owatari, Marco Shizuo

Brazil, July 2023
